# Trajectory Modeling by Distributed Gaussian Processes in Multiagent Systems

**DOI:** 10.3390/s22207887

**Published:** 2022-10-17

**Authors:** Dongjin Xin, Lingfeng Shi

**Affiliations:** School of Electronic Engineering, Xidian University, Xi’an 710071, China

**Keywords:** trajectory modeling, data-driven approach, distributed Gaussian processes, Lyapunov function, model predictive control (MPC)

## Abstract

This paper considers trajectory a modeling problem for a multi-agent system by using the Gaussian processes. The Gaussian process, as the typical data-driven method, is well suited to characterize the model uncertainties and perturbations in a complex environment. To address model uncertainties and noises disturbances, a distributed Gaussian process is proposed to characterize the system model by using local information exchange among neighboring agents, in which a number of agents cooperate without central coordination to estimate a common Gaussian process function based on local measurements and datum received from neighbors. In addition, both the continuous-time system model and the discrete-time system model are considered, in which we design a control Lyapunov function to learn the continuous-time model, and a distributed model predictive control-based approach is used to learn the discrete-time model. Furthermore, we apply a Kullback–Leibler average consensus fusion algorithm to fuse the local prediction results (mean and variance) of the desired Gaussian process. The performance of the proposed distributed Gaussian process is analyzed and is verified by two trajectory tracking examples.

## 1. Introduction

Trajectory tracking is a common problem in control and robotics, and its generation systems represent a large class of dynamical physical models. In the past few decades, various control schemes have been investigated and modeled, and most of them can be considered as a subset of computed torque control laws [1]. Generally speaking, in order to track trajectories, one needs to know the system model, such as the kinematic model, observation model, and motion model [2]. However, in many practical applications, one usually cannot obtain the model information/knowledge, or the system model is dynamical and is difficult to characterize. The system model is often filled with a high degree of uncertainty, nonlinearity, and dependency, which makes it difficult to model accurately. Therefore, traditional modeling methods are no longer suitable for the actual dynamical environment [3]. More recently, data-driven approaches are getting more and more attention in many fields, such as the control and machine learning communities [4,5]. Since data-driven methods can train the system model with high efficiency and precision, they have become the most popular choice for system modeling [6,7]. In particular, the Gaussian process (GP) is the most representative one, and it has been successfully applied to many fields.

A research frontier in the realm of GP is the trajectory modeling issue. Due to its capability to tackle complex perturbations, uncertainties, dependencies, and nonlinearities, GP is becoming a popular choice in various systems, such as in solar power forecasting [8], permanent magnet spherical motors [9], iterative learning control [10], and in swarm kinematic model [11]. In particular, GP has been proven to be effective in improving the learning accuracy and the learning effectiveness of uncertainties and dependencies in low data regimes [12]. More recently, a non-parametric Gaussian process (GP) was proposed for modeling with quantifiable uncertainty and nonlinearity [13,14] based on implicit variance trade-off [15,16]. This bridges the system modeling and data-driven methods. However, computational burden and hardware requirements make GP impractical for big data sets. Furthermore, the high cost of GP also severely hinders the application to an actual physical system. The engineering community has acknowledged these limitations and has attempted to address the problem. Since one can decompose the learning process into a part for a solution, this inspires one to address it in a distributed manner. Accordingly, a distributed GP is an urgent need [17,18].

Generally speaking, the processing is called distributed manner if it is carried out by a cooperative strategy among nodes without central coordination [19]. The distributed method aims at minimizing the amount of computation and communication required by each node as well as making these requirements scalable in the number of nodes [20]. Distributed methods are available for parameter estimation [14], Kalman filtering [21], control [22], optimization [23], learning [13], etc. A major division among distributed methods is based on whether all nodes estimate the full system state [24] or whether each node only estimates a subset of the state variables [25]. The challenge consists in how to execute the update and fusion step in a distributed manner. Existing fusion strategies are usually from the perspective of state estimation and estimation error covariance. Since GP is indeed a Gaussian probability density function (PDF), the trajectory model constructed by GP requires us to consider fusion strategy from the view of PDF [26]. Therefore, this paper is targeted to design a novel GP fusion strategy for multi-agent systems. Generally, the strategy is organized as follows: after obtaining the local predicted results of GP, we perform a fusion of the Kullback–Leibler average consensus on local predictions of GP among neighbors. The distributed GP model can then be developed and successfully applied in large-scale multi-agent systems.

### 1.1. Related Works

Gaussian process-based modeling and based trajectory tracking have been widely investigated and applied over the past two decades. In the first place, most focus on the centralized GP and the multi-input–output GP. In addition, they are developed based on the need for engineering applications in learning and control fields such as GP-based tracking control, state space model learning, and their applications to trajectory tracking. For example, Beckers et al. studied the stable Gaussian process-based tracking control of Lagrangian systems [1]. Umlauft et al. learned stable Gaussian process state space models [27], while Mohammad et al. learned stable nonlinear dynamical systems with Gaussian mixture models [5]. In addition, Pushpak et al. designed control barrier functions for unknown nonlinear systems using Gaussian processes [28]. Umlauft et al. considered human motion tracking with stable Gaussian process state space models in ref. [29] and proposed an uncertainty-based control Lyapunov approach for control-affine systems modeled by the Gaussian process [30]. They also calculated uniform error bounds of Gaussian process regression with application to safe control [31]. Even Gaussian process-based trajectory tracking and control are becoming research hotspots; they focus mainly on one agent and are seldom involved in multi-agent systems. In the second place, distributed and centralized GPs are flourishing in solving data-driven learning algorithms for multi-agent systems. Generally speaking, the main research results are organized as follows: (1) For contributions of models and theories, the unknown map function was modeled and characterized as a GP but with zero-mean assumption, and a distributed parameter and non-parameter Gaussian regression was proposed by using Karhunen–Loeve expansion in refs. [13,14]. To scale GP to large datum, Deisenroth et al. introduced a robust Bayesian committee machine, a practical and scalable product-of-experts model for large-scale distributed GP regression [32]. To address the hyperparameter optimization problem in big data processing, Xie et al. proposed an alternative distributed GP hyperparameter optimization scheme using the efficient proximal alternating direction method of multipliers [33]. Multiple-task GP was studied in ref. [34], while multi-out regression by GP was studied in ref. [35]. Both of them were centralized approaches and could not be extended to a large-scale problem. GP networks were flexible and effective to be used in multi-output regression by combining with variational inference and distributed variational inference in ref. [36], which involved applications to settle non-linear dimension reduction and regression, and provided a powerful tool to address uncertainty and over-fitting problems. (2) For engineering applications, Nerurkar et al. [37] presented a distributed conjugate gradient algorithm for cooperative localization. Franceschelli and Gasparri [38] presented a distributed gossip-based approach to address the pose estimation problem. Cunnigham et al. [39] developed an approach for robot smoothing and mapping by using Gaussian elimination. Distributed localization from distance measurements is studied in [40]. The distributed position estimation was considered in [41]. Distributed rotation estimation algorithm was developed in various engineering [42,43,44]. Distributed Gauss–Seidel algorithm was studied in [45]. GP for data learning in robotic control was considered in [46]. (3) For trajectory tracking in a multi-agent system, an efficient algorithm was presented in ref. [47] to generate trajectory. Gaussian mixture models were used to learn stable trajectory in ref. [5]. The centralized GP for human motion tracking was studied [48]. (4) For distributed model predictive control (MPC), an overview and future research opportunities were discussed in ref. [49]. A cooperative distributed model predictive control for nonlinear systems was studied in [50], for tracking was studied in ref. [51], for linear systems was studied in ref. [52], and for event-based communication and parallel optimization, it was developed in ref. [53]. Additionally, non-cooperative distributed model predictive control was investigated in ref. [54]. More recently, explicit distributed and localized model predictive control via system-level synthesis was investigated in refs. [55,56] and was applied to the trajectory generation of a multi-agent system in ref. [57]. In short, the study of distributed GP is scarce, especially in the trajectory modeling problem.

### 1.2. Contributions

More recently, GP was widely used to model tracking systems and applied to track the target in a real-world environment, such as speed racing (quadrotors) [58], trajectory tracking for wheeled mobile robots [59], 3D people tracking [60], and Simultaneous Localization and Mapping (SLAM) [61,62]. However, these applications focus on one agent, which ignores the advantages of multi-agent systems. After surveying these related references, we find that the trajectory tracking problem is mainly solved by control methods, not data-driven methods, and we focus on one agent, not a multi-agent collaboration. In addition, these existing GP-based learning algorithms are limited by training manners. Motivated by the above discussion, we investigate the distributed GP to learn the trajectory system model in this paper. More specifically, the main contributions of the paper are four-fold. (1) Compared with GP in Lagrangian systems [1,58], this paper considers a general state-space model for both discrete-time and continuous-time. (2) Compared with centralized GP in the state space model [27,28,29,30], PD control and model predictive control are combined together with GP to achieve tracking system modeling and estimating, which can make the estimation error globally uniformly bounded. (3) Compared with existing multiple GP-based centralized approaches, such as collaborative GP [32,33,34,35,63,64], this paper achieves a distributed GP manner to estimate the state, and we apply Kullback–Leibler (KL) average consensus to fuse local training results of GPs, which is different with the Wasserstein metric for measuring GP [65]. (4) Compared with the centralized GP without giving the performance bound [32,33,34,35,63,64] or only providing Kullback–Leibler average analysis [26], this paper analyzes the probabilistically globally ultimate bound of distributed GP.

### 1.3. Paper Structure

The remainder of the paper is organized as follows. Section 2 introduces some preliminaries, including notations, graph theory, Gaussian process, and Kullback–Leibler average consensus. Section 3 states the considered systems. Section 4 designs the local control strategy and proposes a Kullback–Leibler (KL) average consensus to fuse the local predictions of GP. Section 5 provides two tracking experiments. Finally, Section 6 concludes the paper.

## 2. Preliminaries

### 2.1. Notation

Throughout the paper, vectors and vector-valued functions are denoted with bold characters. Matrices are described with capital letters. trace(·), log(·), det(·), 〈·,·〉, ∥·∥, ⊕, ⊙, N(·,·), and GP(·,·) denote, respectively, the trace operation, the logarithm operation, the determinate operation, the inner product, the 2-norm of a matrix or vector, the addition operation of probability density functions (PDFs), the multiplication operation of PDFs, a Gaussian distribution, and a Gaussian process. Moreover, $\dot{x}$, $\ddot{x}$, $\hat{x}$, and $\bar{x}$ denote, respectively, the first-order differential operation on x, the second-order differential operation, the prediction, and the mean operation of x. In addition, $D_{\mathrm{KL}}(p∥q)$, E V, $I_{n}$ denote, respectively, the KL divergence/distance between probabilities p and q, the expectation operation, the variance operation, and an n-by-n identity matrix. In addition, $\frac{dx}{du}$, $\frac{\partial x}{\partial u}$, and O(·) denote, respectively, the derivative operation, the partial derivative operation, and the complexity.

### 2.2. Graph Theory

A graph is defined as G=(N,ε); where N is a set of nodes and ε⊆N×N a set of edges. In particular, graph G is undirected iff (u,v)∈ε⇔(v,u)∈ε for all u,v∈N. The order is |N| and the size of G is |ε|. Further, let $N^{i}=\left\{j\in N:\left(j,i\right)\right\}\in \varepsilon$ denote the set of neighbors for node i.

### 2.3. Gaussian Process

**Definition** **1.**([66]). *A Gaussian process is a collection of random variables, any finite number of which have a joint Gaussian distribution.*

A Gaussian process is completely specified by its mean function and covariance function. We define mean function m(x):x∈χ→ℝ and the covariance function (kernel) k(x,x′):χ×χ→ℝ of a real process f(x) as m(x)=E(f(x)), k(x,x′)=E[(f(x)−x)(f(x′)−x′)], and denote the Gaussian process as f(x)∼GP(m(x),k(x,x′)). When f:χ→ℝn is an n-dimensional map, the GP can be denoted by fj(x)∼GP(m(x),k(x,x′)), j∈{1,…,n}.

Then, the Gaussian process can be organized as [29]
(1)$$f\left(x\right)=\{\begin{array}{c} f_{1}\left(x\right)∼GP\left(m_{1}\left(x\right),k_{1}\left(x,x^{\prime }\right)\right)\\ \vdots \\ f_{n}\left(x\right)∼GP\left(m_{n}\left(x\right),k_{n}\left(x,x^{\prime }\right)\right) \end{array}.$$

In addition, the covariance function (kernel) measures similarity between any two states/variables $x,x^{\prime }\in \chi$ and the common kernel functions include the linear kernel, squared-exponential (SE) kernel, the polynomial kernel, the Gaussian kernel, and the Matèrn kernel.

**Assumption** **1***. Suppose the measurement equation is*y=f(x)+ϵ*, where*y∈ℝm *is the observed vector,*x∈X⊂ℝn *is the state vector defined in a compact set*X*, and*ϵ*is the measurement noise obeying a Gaussian distribution with zero mean and variance*σ2In *(denoted by*ϵ∼N(0,σ2In)*). In addition,*f*is the unknown mapping function (*f:χ→ℝn*) and is assumed to be a GP (**denoted by*f∼GP(0,kθ(x,x′))*). Here*kθ(x,x′)*is a kernel function with respect to hyper**-parameters*θ*,*KXX=kθ(X,X)*denotes the covariance matrix of set*X*, and*KxX=kθ(x,X) *denotes the covariance matrix between*x *and*X.

Generally speaking, given a training set D=(X,y), (input X and output y) [29], the log-likelihood can be computed by
(2)$$\begin{array}{l} \log p\left(y|X,\theta \right)=-\frac{1}{2}y^{T}{\left(K_{XX}+\sigma^{2}I_{n}\right)}^{-1}y\\ \;-\frac{1}{2}\log|K_{XX}+\sigma^{2}I_{n}|-\frac{n}{2}\log2\pi . \end{array}$$

Then, when a new input $x^{*}$ is introduced, the posterior prediction of the Gaussian process [29] is $p\left(f\left(x^{*}\right)\right)=N\left(\mu \left(x^{*}\right),\sum \left(x^{*}\right)\right)$, where
(3)$$\begin{array}{c} \mu \left(x^{*}\right)=K_{x^{*}X}{\left(K_{XX}+\sigma^{2}I_{n}\right)}^{-1}y,\\ \;\sum \left(x^{*}\right)=K_{x^{*}X}-K_{x^{*}X}{\left(K_{XX}+\sigma^{2}I_{n}\right)}^{-1}K_{Xx^{*}}. \end{array}$$

To summarize, the likelihood maximization of (2) is performed to compute gradients for training, and the mean and covariance functions (3) are used for fast predictions.

More specifically, given an arbitrary new testing input x*∈χ conditioning a dataset D described above, the prediction response y* is jointly Gaussian distributed with the training set, which is given by
(4)[yj∗yj]∼N([mj(x∗)mj],[kj∗kjTkjKj+σ2I]),yj=[yj(1)⋯yj(N)]T∈ℝn,mj=[mj(x(1))⋯mj(x(N))]T∈ℝn,kj∗=kj(x∗,x∗)∈ℝ,kj=[kj(x(1),x∗)⋯kj(x(N),x∗)]T,Kj=[kj(x(1),x(1))⋯kj(x(1),x(N))⋮⋱⋮kj(x(N),x(1))⋯kj(x(N),x(N))]∈ℝn×n.

For j = 1,…, *n*, the posterior distribution corresponding to fj(·) at x^∗^ yields a Gaussian distribution with mean function and covariance function as
(5)$$\begin{array}{l} E\left[y_{j}^{*}|D,x^{*}\right]=m_{j}\left(x^{*}\right)+k_{j}^{T}{\left(K_{j}+\sigma^{2}I_{n}\right)}^{-1}\left(y_{j}-m_{j}\right),\\ V\left[y_{j}^{*}|D,x^{*}\right]=k_{j}^{*}-k_{j}^{T}{\left(K_{j}+\sigma^{2}I_{n}\right)}^{-1}k_{j}. \end{array}$$

Furthermore, in order to learn the hyper-parameter $\theta_{j}$ given a chosen kernel, we can use the maximum likelihood function based on Bayes’ rules as
(6)$$\underset{\theta_{j}}{\max}\log p\left(y_{j}|x_{\left(1:n\right)},\theta_{j}\right)=\underset{\theta_{j}}{\max}\left(-\frac{1}{2}y_{j}^{T}K_{j}^{-1}y_{j}-\frac{1}{2}\log detK_{j}-\frac{n}{2}\log\left(2\pi \right)\right).$$
which can be solved by gradient-based approaches [29].

### 2.4. Kullback–Leibler Average Consensus Algorithm

This section introduces the consensus/fusion algorithm of GPs. A Gaussian process is a Gaussian probability density function over mean function and covariance function. Therefore, the fusion of GPs is indeed the fusion of probabilities. It raises a problem: How to achieve consensus/fusion of probabilities among multiple agents?

Before proceeding on, we first introduce some definitions.

**Definition** **2.**(Probability space [26]). Let P≜{p(•):ℝn→ℝ such that ∫ℝnp(x)dx=1 and p(x)≥0,∀x∈ℝn}
*denote the set of probabilities (PDFs) over*
ℝn
*and let*
$p^{i}\left(\cdot \right)\in P\left(i\in N\right)$
*denote the local probability/PDF of agent*
i.

**Definition** **3.***(Kullback–Leibler divergence* [26]). *In statistics, the Kullback–Leibler divergence,*
$D_{KL}\left(p∥q\right)$
*(also called relative entropy), is a statistical distance: a measure of the probability distribution p(·) is different from the probability distribution q(·), which is defined as (for distributions*
p(·)
*and*
q(·)
*of a continuous random variable*
x*)*
(7)$$D_{KL}\left(p∥q\right)=\int p\left(x\right)\log\frac{p\left(x\right)}{q\left(x\right)}dx.$$

**Definition** **4.***(Probabilistic operation* [26]*). Define the plus*
⊕
*and multiplicative*
⊙
*operators over probabilities (*
p(·)
*and*
q(·)
*) for a variable (*
x
*) and a real constant as a*


(8)
$$\begin{array}{c} p\left(x\right)⊕q\left(x\right)≜\frac{p\left(x\right)q\left(x\right)}{\int p\left(x\right)q\left(x\right)dx},\\ a⊙p\left(x\right)≜\frac{{\left[p\left(x\right)\right]}^{a}}{\int {\left[p\left(x\right)\right]}^{a}dx}. \end{array}$$


Then, we attempt to find a Kullback–Leibler average consensus/fusion algorithm over probabilities obtained by multiple agents.

First, according to [26], Kullback–Leibler average (KLA) is to average over probabilities. Motivated by this, we define the weighted KLA ($\bar{p}$) among the probabilities ${\left\{p^{i}\right\}}_{i=1}^{N}$ as
(9)$$\bar{p}=\;\underset{p\in P}{arg\inf}\sum_{i\in N}a^{i}D_{\mathrm{KL}}(p∥p^{i}),$$
where $a^{i}\geq 0$ denotes the weight of agent i and satisfies $\sum_{i\in N}\;a^{i}=1$.

Then, the average consensus/fusion problem is to achieve
(10)$$\underset{l\to \infty }{\lim}p_{l}^{i}=\bar{p},$$
for all agents i∈N, where l is the consensus step and $\bar{p}$ represents the asymptotic KLA with uniform weights.

Second, we attempt to find the solution of the average consensus $\bar{p}$ in (9). Based on [26], the solution is
(11)$$\bar{p}\left(x\right)=\frac{\prod_{i\in N}{\left[p^{i}\left(x\right)\right]}^{a^{i}}}{\int \prod_{i\in N}{\left[p^{i}\left(x\right)\right]}^{a^{i}}dx}≐\underset{i\in N}{⊕}\left(a^{i}⊙p^{i}\left(x\right)\right),$$
with $a^{i}=1/\left|N\right|$. In addition, the local consensus of agent i at the *l*-th consensus step can be obtained by
(12)$$p_{l}^{i}\left(x\right)=\underset{i\in N}{⊕}\left(a^{i,j}⊙p_{l-1}^{j}\left(x\right)\right),\forall i\in N,$$
where $a^{ij}$ is the consensus weight satisfying $a^{ij}\geq 0$, $\sum_{i\in N}a^{i,j}=1$ and $a^{ij}$ represents the (i,j)-th component of the consensus matrix A (if $j\notin N^{i}\;a^{i,j}=0$). Therefore, when the l-th iteration of the consensus algorithm is initialized by $p_{0}^{i}\left(\cdot \right)=p^{i}\left(\cdot \right)$, we can finally obtain the consensus as


(13)
$$p_{l}^{i}\left(x\right)=\underset{i\in N}{⊕}\left(a_{l}^{i,j}⊙p^{j}\left(x\right)\right),\forall i\in N$$


Third, for special Gaussian case, the local probability pi(·) takes the form as
(14)pi(x)=N(x;μi,∑i)≜1det(2π∑i)e−12(x−μi)T(∑i)−1(x−μi),
where μi∈ℝn and ∑i∈ℝn×n denote the mean vector and the covariance matrix, respectively. In view of this case, the KLA can be directly obtained by operating the means and the covariances instead of probabilities. The following lemma states the KLA on Gaussian distributions.

**Lemma** **1.**([26]). *Given*
N
*Gaussian distributions*
$\left(p^{i}\left(x\right),i=1,\ldots ,N\right)$
*defined in (14), with corresponding weigh *
$a^{i}$
*, then the weighted KLA*
$\bar{p}\left(\cdot \right)=N\left(\cdot ;\bar{\mu ,}\bar{\sum }\right)$
*can be calculated by directly fusing the means *
$\mu^{i}$
*and the covariance matrices*
$\sum^{i}$
*as*


(15)
$$\begin{array}{c} {\bar{\sum }}^{-1}=\sum_{i=1}^{N}a^{i}{\left(\sum^{i}\right)}^{-1},\\ {\bar{\sum }}^{-1}\bar{\mu }=\sum_{i=1}^{N}a^{i}{\left(\sum^{i}\right)}^{-1}\mu^{i}. \end{array}$$


Lemma 1 indicates that the consensus/fusion of Gaussian probabilities can directly operate their means and covariance matrices. Note that a Gaussian process is indeed a Gaussian probability. Therefore, the KLA consensus/fusion on GPs can be directly obtained by fusing the mean functions and the covariance functions.

### 2.5. Uniform Error Bounds

This section analyzes the probabilistic uniform error bounds.

**Definition** **5.***(Probabilistic uniform error bound* [31]). ∀x∈X*, if there exists a function*
η(x)
*such that*
∥μ(x)−f(x)∥≤η(x)*, then, on a compact set*
X⊂ℝn*, GP has a uniformly bounded error. A probabilistic uniform error bound is one that holds with a probability of at least*
1−δ
*for any*
δ∈(0,1).

**Definition** **6.***(Lipschitz**constant of the kernel* [64]). *The Lipschitz constant of a differentiable covariance kernel*
k(·,·)
*is*



(16)
$$L_{k}:=\underset{x,x^{\prime }\in X}{max}\left\|{\left[\begin{array}{ccc} \frac{\partial k\left(x,x^{\prime }\right)}{\partial x_{1}} & \cdots & \frac{\partial k\left(x,x^{\prime }\right)}{\partial x_{n}} \end{array}\right]}^{T}\right\|.$$



Next, we show that the posterior prediction (3) of GP is continuous. Given the continuous unknown f with Lipschitz constant Lf and the Lipschitz continuous kernel k with Lipschitz constant $L_{k}$, we then have the following theorem.

**Theorem** **1.**([31]). *Given a GP defined by the continuous covariance kernel function*
k
*with Lipschitz constant*
$L_{k}$
*, a continuous unknown map*
f
*with Lipschitz constant*
$L_{f}$
*and measurements satisfying Assumption 1. Then, the posterior predictions*
(μ(·) and ∑(·))
*of the GP conditioning on the training date set*
$D={\left\{x_{t},y_{t}^{i}\right\}}_{t=1,\ldots ,N}^{i=1,\ldots ,N}$
*are continuous with Lipschitz constant*
$L_{\mu }$
*and modulus of continuity*
$\omega_{\sum }$
*such that*




(17)
$$\begin{array}{c} L_{\mu }\leq L_{k}\sqrt{N}\left|{\left(K+\sigma^{2}I\right)}^{-1}y\right|,\\ \omega_{\sum \left(\tau \right)}\leq \sqrt{2\tau L_{k}\left(1+N\left|{\left(K+\sigma^{2}I\right)}^{-1}\right|\underset{x,x^{\prime }\in X}{max}k\left(x,x^{\prime }\right)\right)}. \end{array}$$

*for any*

τ∈ℝ+

*and*

δ∈(0,1)

*with*




(18)
$$\begin{array}{l} \beta \left(\tau \right)=2\log\left(\frac{M\left(\tau ,X\right)}{\delta }\right),\\ \gamma \left(\tau \right)=\left(L_{\mu }+L_{f}\right)\tau +\sqrt{\beta \left(\tau \right)}\omega_{\sum \left(\tau \right)}. \end{array}$$




*In addition,*

∀x∈X

*, it follows that*

(19)
$$p\left(\left\|f\left(x\right)\right\|-\mu \left(x\right)\leq \gamma \left(\tau \right)+\sqrt{\beta \left(\tau \right)\sum \left(x\right)}\right)\geq 1-\delta .$$



**Proof** **of** **Theorem** **1**.The proof is given in [App appA-sensors-22-07887]. □

#### Asymptotic Analysis

The asymptotic analysis of the error bound (19) in the limit N→∞ is organized as the following theorem.

**Theorem** **2.**([31]). *Given a GP defined by the continuous covariance kernel function*
k
*with Lipschitz constant*
$L_{k}$*, and an infinite data response of measurements*
$\left(X_{t},y_{t}^{i}\right)$
*of the continuous unknown map*
f *with Lipschitz constant*
$L_{f}$
*and the maximum absolute value*
$∥f^{max}∥$*. The first N measurements inform the posterior predictions of the GP as*
(μ(·) and ∑(·))*. If there exists a*
ϵ>0
*such that*
$\sum \left(x\right)\in O\left(\log{\left(N\right)}^{-\frac{1}{2}-\epsilon }\right)$*,*
∀x∈X *for any*
δ∈(0,1)*, it follows that*
(20)$$p\left(\underset{x\in X}{\sup}\left\|f(x)-\mu_{N}(x)\right\|\in O\left(\log{\left(N\right)}^{-\frac{1}{2}-\epsilon }\right)\right)\geq 1-\delta .$$

**Proof** **of** **Theorem** **2**.The proof is given in [App appB-sensors-22-07887]. □

## 3. Problem Formulation

The trajectories generate from a continuous dynamical system
(21)x˙=f(x,u)+w,
where x∈χ⊂ℝn in a compact set, χ denotes the state (location), u∈U⊆ℝn denotes the control input, ω denotes the process noise with w∼N(0,σω2I) and the initial state is x(0)=x0. We have N agents/sensors connected with a network to acquire measurements (location or velocity). In particular, sensor i measures
(22)yji=fi(x)+ϵji,
where yji is the observed vector of sensor i(i=1,…,N) at the j-th step (j=1,…,N), ϵji is the measurement noise with ϵji∼N(0,σ2I).

Suppose that a training data set D of trajectories is given. D contains the state (current location) and the measurement, which is denoted by D={xi,yji}i=1…Nj=1…N. The nonlinear map function f:χ→ℝn is unknown and is assumed to be a Gaussian process. In addition, the following assumption is satisfied.

**Assumption** **2.**f(x)*Suppose the measurement is Lipschitz continuous and has a bounded RKHS (reproducing kernel Hilbert space) norm with respect to fixed common kernel*k*,*${∥f∥}_{k}=\sqrt{f,f_{k}}<\infty$.

The objective is to find an estimated $\hat{f}$ of f, for which the output trajectory x tracks the desired trajectory $x_{d}={\left[\begin{array}{cc} x_{d} & {\dot{x}}_{d} \end{array}\right]}^{T}$ such that the tracking error $e=x-x_{d}={\left[\begin{array}{cc} e_{1} & e_{2} \end{array}\right]}^{T}$ vanishes over time, i.e., $\underset{t\to \infty }{lim}∥e∥=0$. l. Since the noises ω and $\epsilon_{j}^{i}$ and the uncertain dynamics affect the system and control, we use multiple agents to eliminate the influence of stochastic uncertainty, i.e., given local ${\hat{f}}^{i}$, the goal is also to fuse them and to find a fused/consensus $\bar{f}$ such that the uncertainty also vanishes over time.

## 4. Control Design and Analysis

Classical control uses static feedback gains. Low feedback gains are designed to avoid saturation of the actuators and good noise suppression. However, the considered unknown dynamics require a more minimal feedback gain to keep the tracking error under a defined limit. After performing a training procedure, we use the mean function of GP to adapt the gains. For this purpose, the uncertainty of the GP and multiple agents are employed to scale the feedback gains.

Before proceeding on, the following natural assumptions and lemmas are given.

**Assumption** **3.***The desired trajectory*$x_{d}\left(t\right)$*is bounded by*$∥x_{d}∥={∥[\begin{array}{cc} x_{d} & {\dot{x}}_{d} \end{array}]}^{T}∥\leq q_{d}={\left[\begin{array}{cc} q_{d} & {\dot{q}}_{d} \end{array}\right]}^{T}$.

**Lemma** **2.**([67]) *If there exist a positive constant*
b∈ℝ+
*such that*
∀a∈ℝ+*, if there exists a function*
T=T(a,b)
*satisfying*
$∥x(t_{0})∥\leq a$
*, then we have that*
$\forall t\geq t_{0}+T,∥x\left(t\right)∥\leq b$
*, i.e., the trajectory*
x(t)
*of the dynamics (21) is globally ultimately bounded.*

**Lemma** **3.**([67]) *If there exists a Lyapunov function*
V
*such that*
$\dot{V}\left(x\right)<0$
*for all*
x∈X\B*, the dynamical system*
$\dot{x}=f\left(x,u\right)$
*is globally ultimately bounded to a set*
B⊂X.

Next, we design the controller and the control law such that stability and high-performance tracking are achieved. The controller is designed as
(23)$$u=-\hat{f}\left(x\right)+\rho ,$$
where $\hat{f}$ is the model estimation of nonlinear dynamics f and $\hat{f}$ is obtained by utilizing the posterior mean function ${µ}_{N}$ of GP trained by the data set D, and ρ is the control law.

In addition, ρ is designed as a Proportional–Derivative (PD) type controller
(24)ρ=x¨d−kdr−kpe2,
where r=kpe1+e2 is the filtered state with r˙=f(x)−f^(x)−kcr, kp∈ℝ+, and the control gain kp∈ℝ+.

Given the above controller, one needs to verify the effectiveness of the model estimation $\hat{f}$ and the choices of the parameters $k_{d}$ and $k_{p}$. The following theorem states the control law with guaranteed boundedness of the tracking error.

**Theorem** **3.***Consider the system (23), where f satisfies Assumption 2 and admits a Lipschitz constant*$L_{f}$*. If Assumption 3 is satisfied, then the controller with*$\hat{f}={µ}_{N}$*and the control law guarantee that the tracking error is globally ultimately bounded and converges to a ball*$B=\left\{∥e∥\leq \frac{\gamma \left(\tau \right)+\sqrt{\beta \left(\tau \right)\sum_{N}\left(x\right)}}{k_{c}\sqrt{\lambda^{2}+1}}\right\},\forall x\in X$*, where*β*and*γ*are given in Theorem 1, with a probability of at least*1−δ, δ∈(0,1).

**Proof** **of** **Theorem** **3.**The proof is given in [App appC-sensors-22-07887]. □

**Remark** **1.***From Theorem 3, it can be seen that trajectory tracking with high probability is achieved with the proposed GP-based controller. Compared with most existing results where only uniformly ultimate boundedness of the trajectory tracking errors was achieved* [1,68], *the proposed control law ensures high control precision in the presence of the estimation errors from GP.*

**Proof** **of** **Remark** **1**.The proof is given in [App appC-sensors-22-07887]. □

### 4.1. Consensus

The aforementioned control law focuses one agent/sensor. Since the noises ω and $\epsilon_{j}^{i}$ affect the measurements and the dynamical system, the proposed controller will fluctuate for different agents. Furthermore, since the uncertainty of dynamics exists, the proposed controller may also change for different agents. Therefore, this section will fuse them and make them reach a consensus, i.e., given local ${\hat{f}}^{i}$ (${µ}_{N}^{i}$ and $\sum_{N}^{i}$), the goal is to fuse them and to find a fused/consensus $\bar{f}\left({\bar{µ}}_{N}\;\mathrm{and}\;{\bar{\sum }}_{N}\right)$ such that the uncertainty and the disturbance can vanish over time. Obviously, the controller (23) and the control law (24) for different agents can reach a consensus $\bar{f}\left({\bar{µ}}_{N}\;\mathrm{and}\;{\bar{\sum }}_{N}\right)$.

More specially, after training the local ${\hat{f}}^{i}$ by using GP, node i sends the result to its neighbors $N^{i}$. After collecting the training results from neighbors, it performs the following dynamic consensus/fusion step. Given weights $a^{i}$ satisfying $a^{i}\geq 0$ and $\underset{i\in N^{i}}{\sum }a^{i}=1$ based on the Kullback–Leibler average consensus given in Section 2.4, the desired weighted KLA takes the Gaussian form as $\bar{f}\left(\cdot \right)=N\left(\cdot ;\bar{\mu },\bar{\sum }\right)$, in which the fusion of the mean function $\bar{\mu }$ and the fusion of the covariance function $\bar{\sum }$ can be calculated by
(25)$$\begin{array}{c} {\bar{\sum }}^{-1}=\sum_{i=1}^{M}a^{i}{\left(\sum_{N}^{i}\right)}^{-1},\\ {\bar{\sum }}^{-1}\bar{\mu }=\sum_{i=1}^{M}a^{i}{\left(\sum_{N}^{i}\right)}^{-1}\mu_{N}^{i}, \end{array}$$
while the global/centralized fusion using i=1,…,N. The flowchart is given in Figure 1.

After obtaining the consensus mean function, the controllers of different agents can be designed to be a unified controller.

**Remark** **2.**
*The main advantage of the distributed method lies in that local nodes can only receive part of the training data or even missing data. Neighboring nodes can make the prediction faster and keep high accuracy through information interaction and consensus algorithm, which can also avoid processor failure caused by data loss or node/sensor failure.*


### 4.2. GP-Based Model Predictive Control for Discrete-Time System

The above discussion discusses the continuous-time system. Usually, we need to discretize the system in an actual physical system. This section designs the control strategy for discrete-time by using GP-based model predictive control (MPC).

First, the considered system (21) is assumed to be discrete-time and can be modeled by GP, where the control tuple $x_{k}={\left[\begin{array}{cc} x_{k} & u_{k} \end{array}\right]}^{T}$ and the state difference $\delta x_{k}=x_{k+1}-x_{k}$ are, respectively, designed as the training input and the desired target. Given the training date set $D={\left\{x_{t},y=\delta x_{k}\right\}}_{k=1\ldots N}$, according to (4) and (5), at a new training input $x^{*}$, we can obtain the mean function and covariance function as follows
(26)$$\begin{array}{l} E\left[\delta x_{k}|D,x^{*}\right]=k^{T}{\left(K+\sigma^{2}I_{N}\right)}^{-1}y,\\ V\left[\delta x_{k}|D,x^{*}\right]=k^{*}-k^{T}{\left(K+\sigma^{2}I_{N}\right)}^{-1}k, \end{array}$$
where $k^{*}=k\left(x^{*},x^{*}\right)$, $k={\left[\begin{array}{ccc} k\left(x_{\left(1\right)},x^{*}\right) & \ldots & k\left(x_{\left(n\right)},x^{*}\right) \end{array}\right]}^{T}$, and K is defined in (4). Therefore, (26) is given to predict the next step. By using the moment matching approach [46], the mean function and covariance function of the training target at time k can be calculated by
(27)$$\begin{array}{l} \mu_{k}^{\delta }=E\left[E\left[\delta x_{k}\right]\right],\\ \sum_{k}^{\delta }=\left[\begin{array}{ccc} k\left(\delta x_{k_{1}},\delta x_{k_{1}}\right) & \cdots & k\left(\delta x_{k_{n}},\delta x_{k1}\right)\\ \vdots & \ddots & \vdots \\ k\left(\delta x_{k_{1}},\delta x_{k_{n}}\right) & \cdots & k\left(\delta x_{k_{n}},\delta x_{k_{n}}\right) \end{array}\right]. \end{array}$$

At time k+1, the mean and covariance functions are updated as
(28)$$\begin{array}{l} \mu_{k+1}=\mu_{k}+\mu_{k}^{\delta },\\ \sum_{k+1}=\sum_{k}+\sum_{k}^{\delta }+k\left(x_{k},\delta x_{k}\right)+k\left(\delta x_{k},x_{k}\right). \end{array}$$

For more details, please refer to [46].

Then, based on (6), we next attempt to learn the hyper-parameters θ. A distributed GP-based MPC scheme is presented to address this problem. First, we design the objective function as
(29)$$J_{k}=\underset{u}{\min}E\left[V\left(x_{k},u_{k-1}\right)\right],$$
where the cost function is
(30)$$E\left[V\left(x,u\right)\right]=\sum_{l=1}^{L}E\left[{\left(x_{k+l}-p_{k+l}\right)}^{T}Q\left(x_{k+l}-p_{k+l}\right)+u_{k+l-1}^{T}Ru_{k+l-1}\right],$$
where p is the desired trajectory (desired state), Q and R are positive definite weight matrices, and L is the prediction horizon and also the control horizon. According to GP in Section 2, (30) can be rewritten as
(31)$$E\left[V\left(x,u\right)\right]=\sum_{l=1}^{L}E\left[\begin{array}{l} {\left(u_{k+l}-p_{k+l}\right)}^{T}Q\left(u_{k+l}-p_{k+l}\right)\\ +\mathrm{trace}\left(Q\sum_{k+l}\right)+u_{k+l-1}^{T}Ru_{k+l-1} \end{array}\right],$$

Next, to address the optimization problem (29), a gradient-based method is used. Set $F_{l}={\left(u_{k+l}-p_{k+l}\right)}^{T}Q\left(u_{k+l}-p_{k+l}\right)+\mathrm{trace}\left(Q\sum_{k+l}\right)+u_{k+l-1}^{T}Ru_{k+l-1}$ and $E\left[V\left(x,u\right)\right]=\sum_{l=1}^{L}EF_{l}$. Using the chain rule, the gradient can be calculated by
(32)$$\begin{array}{l} \frac{d}{du_{k-1}}E\left[V\left(x_{k},u_{k-1}\right)\right]=\sum_{l=1}^{L}\frac{dF_{l}}{du_{k+l-1}},\\ \frac{dF_{l}}{du_{k+l-1}}=\frac{\partial F_{l}}{\partial u_{k+l}}\frac{\partial u_{k+l}}{\partial u_{k+l-1}}+\frac{\partial F_{l}}{\partial \sum_{k+l}}\frac{\partial \sum_{k+l}}{\partial u_{k+l-1}}+\frac{\partial F_{l}}{\partial u_{k+l-1}}. \end{array}$$
where $\frac{\partial F_{l}}{\partial u_{l}},\frac{\partial F_{l}}{\partial \sum_{l}}\;\mathrm{and}\;\frac{\partial F_{l}}{\partial u_{l-1}}$ are easy to calculate. In addition,
(33)$$\begin{array}{l} \frac{\partial u_{k+l}}{\partial u_{k+l-1}}=\frac{\partial u_{k+l}}{\partial u_{k+l-1}}\frac{\partial u_{k+l-1}}{\partial u_{k+l-1}},\\ \frac{\partial \sum_{k+l}}{\partial u_{k+l-1}}=\frac{\partial \sum_{k+l}}{\partial \sum_{k+l-1}}\frac{\partial \sum_{k+l-1}}{\partial u_{k+l-1}}, \end{array}$$
where $\frac{\partial u_{k+l-1}}{\partial u_{k+l-1}}\;\mathrm{and}\;\frac{\partial \sum_{k+l-1}}{\partial u_{k+l-1}}$ are easy to calculate.

Finally, the gradient-based algorithm is formulated as Algorithm 1.
**Algorithm 1** Gradient-based optimization method ** Input**: learning GP, L, p, Q, and R** Output**: Optimal control u* 1: Initialization: Max iteration number N=1000, threshold ε=10−8, the initialized input u0 and optimal control u*=u0; 2: **for** k=1 to N **do** 3:   **if** E[V]<ε **then** 4:     u*=uk; 5:   **end** Loop; 6:   **else** 7:     Calculate the gradient dE[V(uk)]duk−1 by (32); 8:     Update search step size based on [69]; 9:     Update control uk+l=uk+αldE[V(ul)]dul−1; 10    Go next l→l+1** end**** end** 11: **return** Optimal control u*.

**Remark** **3.**
*Similarly, due to the stochastic uncertainty caused by the noises and model perturbations, we can use multiple agents to address this problem. The consensus/fusion algorithm is given above, which is similar to the continuous system. Therefore, we will not introduce it any more.*


**Remark** **4.**
*The GP has been widely applied in various real-world applications such as quadrotor tracking, 3D people tracking, localization and mapping, and control-based application models. These applications have attracted much attention from engineers and researchers. As for the limitations, in our opinion, the first is that the model needs real-world data to achieve perfect training and application. The second is that the dynamics are Gaussian distributed or Gaussian-approximate distributed.*


## 5. Simulations

To evaluate the performance and to verify the effectiveness of the proposed algorithms, this section provides two trajectory tracking examples, where one is the trajectory tracking of a robotic manipulator and the other one is the trajectory tracking of an unmanned quadrotor. All simulations are conducted on a computer with 2.6 GHz Intel(R) Core(TM) i7-5600U CPU and MATLAB R2015b.

### 5.1. Trajectory Tracking of Robotic Manipulator

First, we consider the trajectory of a Puma 560 robot arm manipulator in x-y-z plane with 6 degrees of freedom (DoFs), which is shown in Figure 2. The Puma 560 robot was designed to have approximately the dimensions and reach of a human worker. It also had a spherical joint at the wrist, just as humans have. Roboticists use like waist, shoulder, elbow, and wrist when describing serial link manipulators. For the Puma, these terms correspond respectively to joints 1, 2, 3, and 4–6, which is shown in Figure 2.

For the considered robot arm, τ1, τ2, and τ3 are the control torques of the motors controlling the joint angles ϕ, θ, ψ. The trajectory of the robotic manipulator can be controlled by these torques. The motion can be described by the following Lagrangian system [1]

(34)$$H(q)\ddot{q}+C(q,\dot{q}+G(q)+\kappa \left(\overset{˘}{q}\right)=\tau ,$$
where q denotes the generalized coordinates with their time derivatives $\dot{q}$, $\ddot{q}$ and τ denotes the generalized input. H is the mass matrix, C is the Coriolis matrix and G is the potential energy matrix. An additional unknown dynamic $\kappa \left(\overset{˘}{q}\right)$(train to obtain its form), which depends on $\overset{˘}{q}={\left[{\ddot{q}}^{T},{\dot{q}}^{T},q^{T}\right]}^{T}$, affects the system as a generalized force, in which one can refer to [2] for details. The process methodology is illustrated in Figure 3.

Tracking error is the error between the actual value of joint angle or velocity with the desired values
(35)$$\begin{array}{l} \tilde{q}=\left[\begin{array}{c} \phi -\phi_{d}\\ \theta -\theta_{d}\\ \psi -\psi_{d} \end{array}\right]=q\left(t\right)-q_{d}\left(t\right),\\ \dot{\tilde{q}}=\dot{q}\left(t\right)-{\dot{q}}_{d}\left(t\right). \end{array}$$

The following controllers are tested. (1) Computed torque (CT) controller: $\tau_{in}=H\left(q\right){\ddot{q}}_{d}+C\left(q,\dot{q}\right){\dot{q}}_{d}+G\left(q\right)-K_{p}\left(\tilde{q}\right)-K_{d}\left(\dot{\tilde{q}}\right)$. The gains for this controller is $K_{p}$ = 50 and $K_{d}$ = 40. (2) The proposed PD controller (24): the composite error is $S=\dot{\tilde{q}}+$λ$\tilde{q}$, a reference velocity ${\dot{q}}_{r}$ is ${\dot{q}}_{r}={\dot{q}}_{d}-\lambda \tilde{q}\tilde{q}=q\left(t\right)-q_{d}\left(t\right)$, and the control torque is $\tau_{in}=H\left(q\right){\dot{q}}_{r}+C\left(q,\dot{q}\right){\dot{q}}_{r}+G\left(q\right)-K_{p}\left(\tilde{q}\right)-K_{d}S$. The gains for this controller are λ = 30 and $K_{d}$ = 20. (3) The adaptive controller: the control torque is $\tau_{in}$=$Y_{0}+Y_{1}{\hat{m}}_{3}+Y_{2}{\hat{I}}_{xx_{3}}+Y_{3}{\hat{I}}_{yy_{3}}+Y_{4}{\hat{I}}_{zz_{3}}-K_{d}S$, where ${\hat{m}}_{3}$, ${\hat{I}}_{xx_{3}}$, ${\hat{I}}_{yy_{3}}$, ${\hat{I}}_{zz_{3}}$ are the unknown parameters. $Y_{0}$ $Y_{1}$ $Y_{2}$ $Y_{3}$ $Y_{4}$ are called the regressor vectors defined as [70]. ${\hat{m}}_{3}\;$=$\;-\left(S^{T}Y_{1}\right)∕\gamma_{1}$, ${\hat{I}}_{xx_{3}\;}$=$\;-\left(S^{T}Y_{2}\right)∕\gamma_{2}$, ${\hat{I}}_{yy_{3}}\;$=$\;-\left(S^{T}Y_{3}\right)∕\gamma_{3}$, ${\hat{I}}_{zz_{3}}\;$=$\;-\left(S^{T}Y_{4}\right)∕\gamma_{4}$, where $\gamma_{1}$, $\gamma_{2}$, $\gamma_{3}$, $\gamma_{4}$ are controller gains, in addition to $\gamma_{1}$ = 50, $\gamma_{2}$ = 30, $\gamma_{3}$ = 20, $\gamma_{4}$ = 50, and $K_{d}$ = 20.

Data. Speeds: 5 s, 10 s, 15 s, and 20 s completion times; 4 paths × 4 speeds with 16 different trajectories; 15 loads (0.2 kg…3.0 kg), various shapes and sizes; 10 agents. Training Data. One desired trajectory common to handling of all loads; one trajectory has no data for any context; sixteen unique training trajectories, one for each load. Test Data. Interpolation data sets for testing on reference trajectory and the unique trajectory for each load. Extrapolation data sets for testing on all trajectories.

From Figure 4, it can be seen that the PD controller action is able to hold the position of the robot arm at the desired joint angles. λ = 30 and $K_{d}$ = 20 are the gains associated with holding the respective positions necessary. The convergence of the plots is achieved in about 10 s. The first couple runs of the robot can be used for tuning the robot, and the robot should have good repeatability after that. In addition, from the position and velocity plots of a computed torque controller (Figure 5) and adaptive controller (Figure 6), it is observed that both controllers are able to achieve convergence of parameters to the desired values. However, the proposed PD torque controller has a quicker convergence time and also has lesser gains compared to the computed torque controller. The error results from Table 1 further verify the effectiveness. Therefore, the proposed PD torque controller is better for the considered application. In addition, we can also obtain that the distributed GP can effectively eliminate the uncertainty and disturbance caused by the system model and the noises.

To further compare with the multiple agent processing methods, these approaches are tested: (1) Independent GP (IGP): model trained independently for each input [6]; (2) Combined GP (CGP): one agent to train GP by combining data across inputs [34]; (3) Proposed distributed GP with BIC (Bayesian Information Criterion) criterion. The training results (interpolation and extrapolation manners) of NMSE (normalized mean square error) with regard to the number of training data points are demonstrated in Figure 7. Note that IGP and CGP, i.e., existing GP, are centralized methods, which are different from the proposed GP, which is a distributed method for training. From Figure 7, the first line displays the training results of the interpolation manner for the three methods. As we can see from it, for joint 1, the proposed distributed GP achieves the best performance for any number of training points. For joint four and joint six, the performance of the proposed GP is close to IGP and also better than CGP when the training points increase. The second line displays the training results of the extrapolation manner for the three methods. As we can see from it, for joint 1, the proposed distributed GP achieves the best performance for small numbers of training points (<500). However, when training points are increased further, the CGP is better than the proposed distributed GP (note that they are very close). For joint four and joint six, the performance of the proposed GP is close to IGP and also better than CGP when the training points increase. To sum up, the proposed distributed GP model can reach the performance of the centralized method and is close to (even outperforms) the existing state-of-the-art centralized-based multiple combined and multi-task methods.

### 5.2. Trajectory Tracking of an Unmanned Quadrotor

This section tests the proposed distributed GP-based model predictive control (GPMPC) of an unmanned quadrotor. The trajectory of an unmanned quadrotor is generated by a discrete-time Euler–Lagrange dynamical system [2]. The goal is to track its positions (X, Y, Z) and Euler angles (ϕ, θ*,* ψ). To compare with the state-of-the-art controllers, the efficient MPC (EMPC) [71] and the efficient nonlinear MPC (ENMPC) [72] are also tested in the simulations. The parameters are selected as L = 5 and Q=R=diag(1,1,1).

In the first scenario, the unmanned quadrotor tracks a “Lorenz” trajectory with Gaussian white noise (zero mean and unit variance), which is shown in Figure 8. To train the system model, we use the efficient MPC design proposed in [71]. One hundred seventy measurements, states, and controls are used to train the GP. The datum from the rotational system is with the range [0, 1], the angle φ is with a range [−1.6, 1.6], and the input is with the range $\left[-4\times 10^{-8},\;7\times 10^{-8}\right]$. The training of GP takes 5 s, and we use 10 agents to train. The values of mean squared error (MSE) trained by GP are very small. The mean squared error (MSE) for the positions is $4.3618\times 10^{4}$ close to the stable GPMPC (GPMPC1) [73] with $4.3498\times 10^{4}$; MSE for the angels is $1.5743\times 10^{8}$ also close to GPMPC1 with $4.3030\times 10^{9}$. This indicates that the proposed distributed GP (GPMPC2) is efficient and well-trained, which is illustrated in Figure 9 (with different confidences). Note that the stable GPMPC (GPMPC1) is the most recent best method at present and is also a method for one agent. Therefore, the training results are very close, which indicates that the proposed distributed GP can achieve a good training performance.

The positions and attitudes tracking results are demonstrated in Figure 10, and the tracking errors are displayed in Figure 11 and Table 2. As we can see from Figure 10 and Figure 11, the proposed distributed GP can learn the system model well and track the trajectories with high precision, which is close to the state-of-the-art controllers. This also indicates that as long as the training sets are introduced, we can track the trajectory without model knowledge (model-free), i.e., the proposed GP can learn the system model well. In addition, as long as multiple agents are introduced, the model uncertainties and noise disturbances can be eliminated and suppressed.

Furthermore, the covariance on positions and attitudes by different GP models (the stable GPMPC (GPMPC1) [69] and the proposed distributed GPMPC (GPMPC2)) is displayed in Figure 12. This indicates that the proposed distributed GP can also reach the performance of the state-of-the-art GP model.

In the second scenario, the unmanned quadrotor tracks an “Elliptical” trajectory with Gaussian white noise (zero mean and unit variance), which is shown in Figure 13. The tracking performance and the tracking errors are shown in Figure 14 and Figure 15 and Table 3. From Figure 13, Figure 14 and Figure 15, we can also see that the proposed distributed GP can learn the trajectory model effectively, which is very close to the desired reference trajectory and is close to the state-of-the-art controllers. The covariance results by GPMPC1 and GPMPC2 are shown in Figure 16, which further verifies the effectiveness of the proposed distributed GP.

## 6. Conclusions

This paper used the Gaussian process to learn the trajectory model, and a distributed GP-based model learning strategy was proposed. For the continuous- and discrete-time system, we, respectively, designed a GP-based PD controller and a GP-based MPC controller to address the problem. To address the uncertainties of the model and the disturbances of the noises, a distributed multiple-agent system was used to train the model. In addition, since data-driven algorithms needed a large number of training sets, the distributed GP model could also be employed to address this problem by using a Kullback–Leibler average consensus fusion criterion.

The proposed GP can solve the actual model-free problem as long as the training data sets are given. Since the considered multi-agent is interconnected and it is only used to eliminate the uncertainties of the model and disturbances of the noises, future research mainly focuses on the efficiency of distributed Gaussian process and the robustness of the multi-agent network. In the future, we will focus on the application deployment of an unmanned aerial vehicle (UAV) and its usage in UAV detection and location. UAV racing is a challenging problem to overcome.

## Figures and Tables

**Figure 1 sensors-22-07887-f001:**
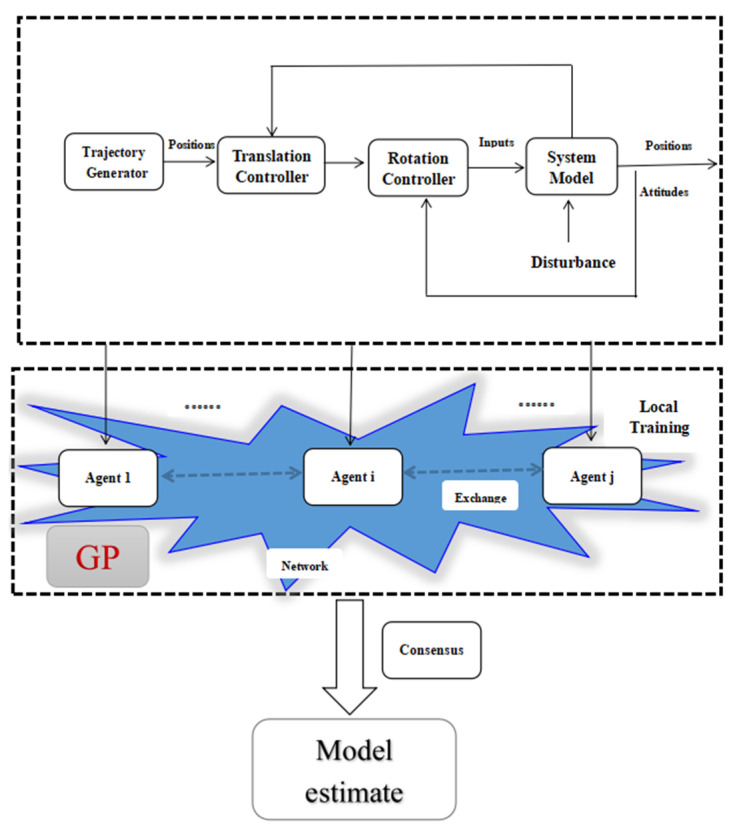
The flowchart of consensus/fusion.

**Figure 2 sensors-22-07887-f002:**
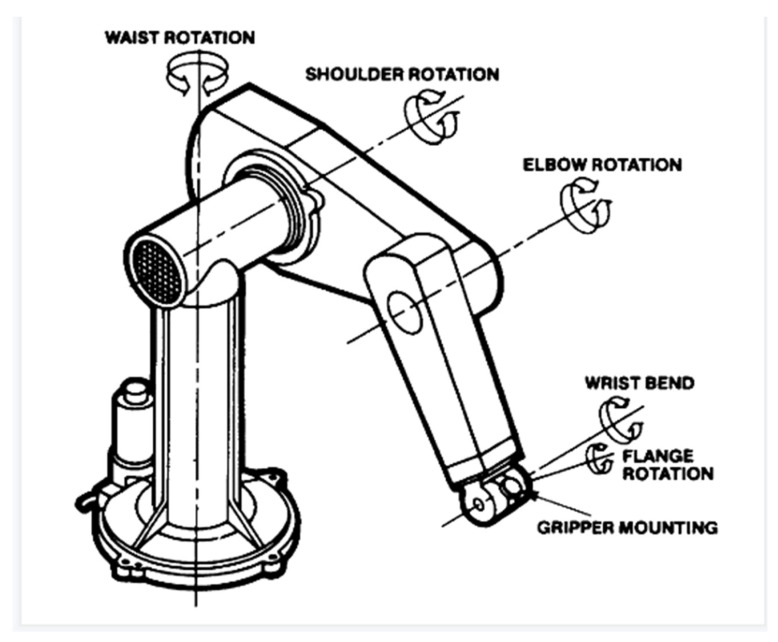
The diagram of the Puma 560 robot arm manipulator (6 DoFs).

**Figure 3 sensors-22-07887-f003:**
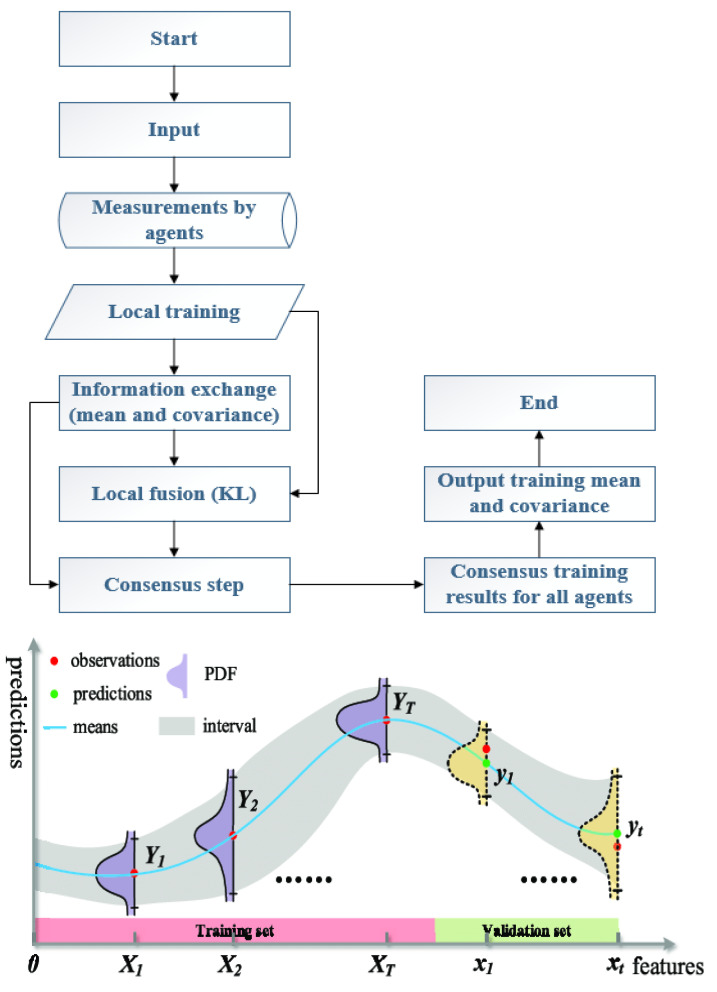
The process methodology and flowchart.

**Figure 4 sensors-22-07887-f004:**
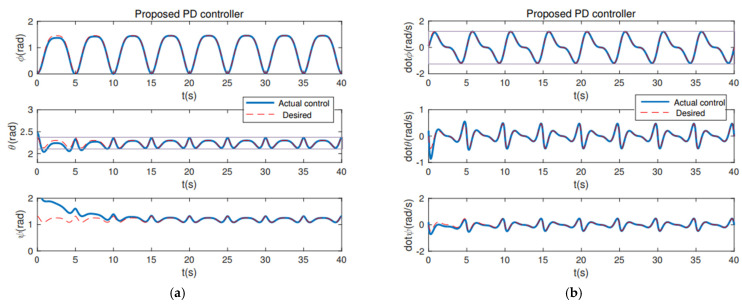
The proposed PD controller: (**a**) Position Plot; (**b**) Velocity Plot.

**Figure 5 sensors-22-07887-f005:**
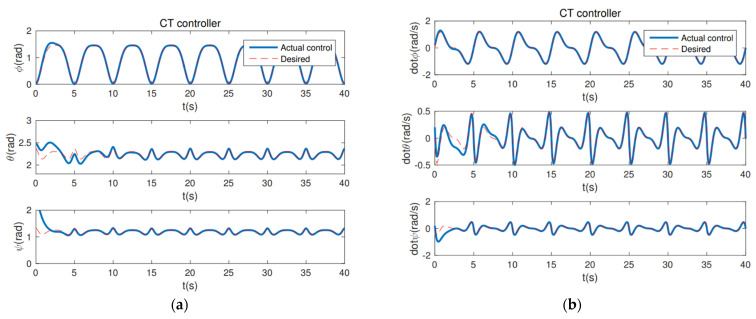
The CT controller: (**a**) Position Plot; (**b**) Velocity Plot.

**Figure 6 sensors-22-07887-f006:**
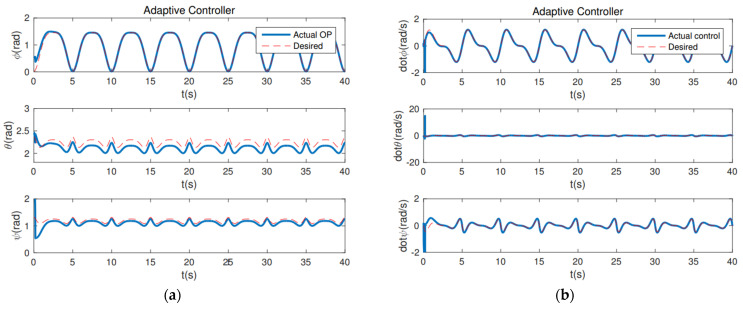
The Adaptive controller: (**a**) Position Plot; (**b**) Velocity Plot.

**Figure 7 sensors-22-07887-f007:**
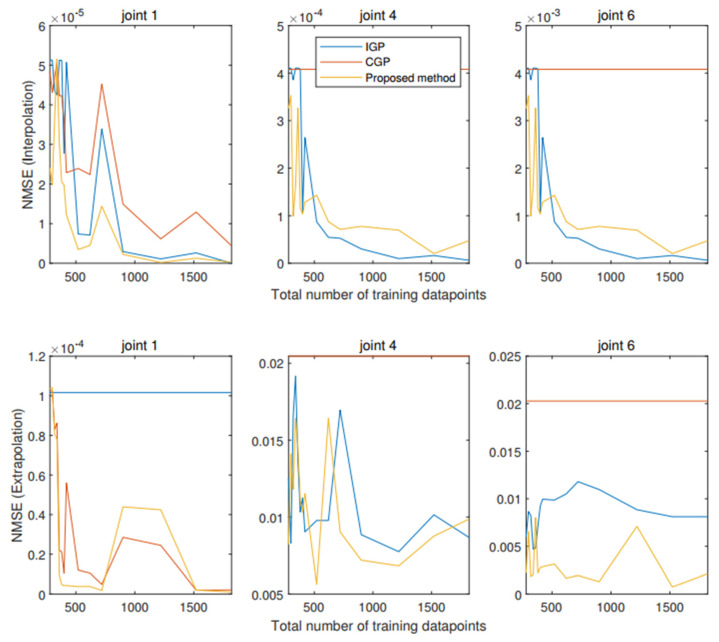
Training results using different methods.

**Figure 8 sensors-22-07887-f008:**
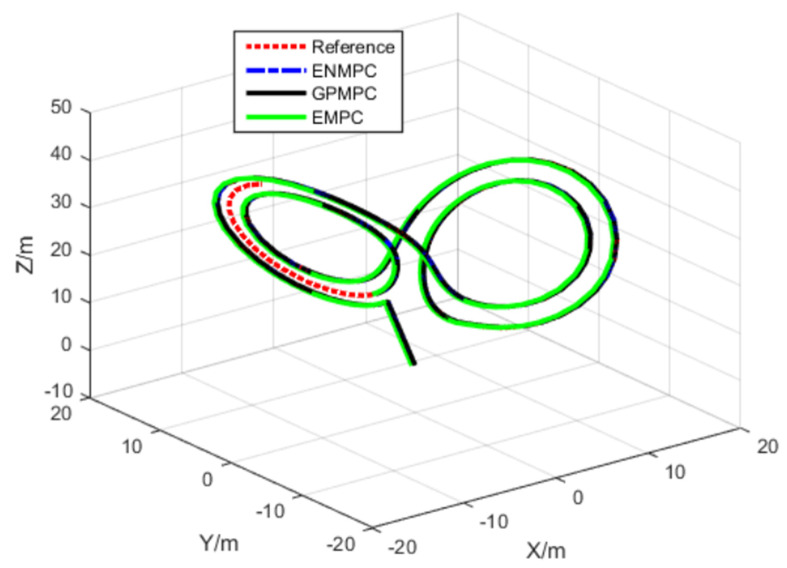
Lorenz trajectory tracking.

**Figure 9 sensors-22-07887-f009:**
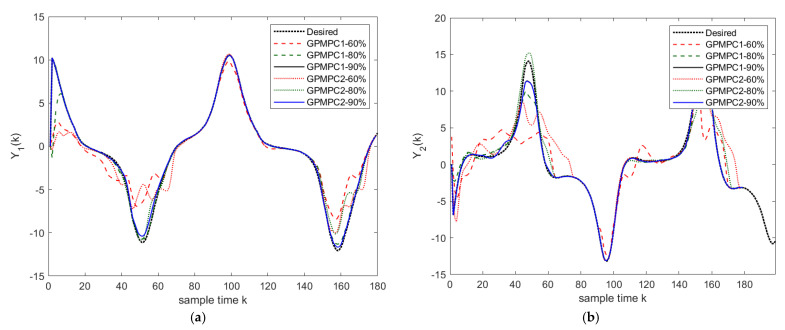
Training performance with 60%, 80%, and 90% confidence: (**a**) $Y_{1}\left(k\right)$; (**b**) $Y_{2}\left(k\right)$.

**Figure 10 sensors-22-07887-f010:**
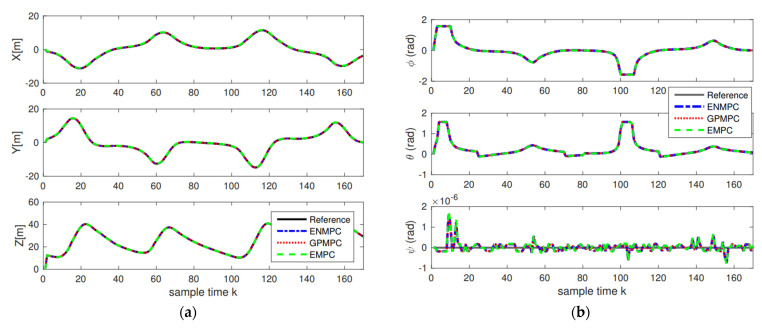
Tracking results: (**a**) Positions tracking; (**b**) Attitudes tracking.

**Figure 11 sensors-22-07887-f011:**
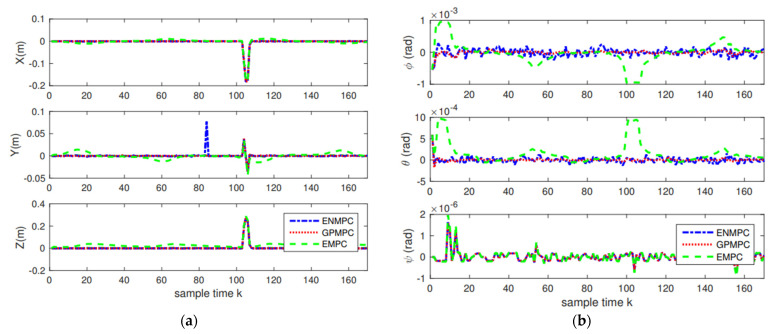
Tracking errors: (**a**) Positions errors; (**b**) Attitudes errors.

**Figure 12 sensors-22-07887-f012:**
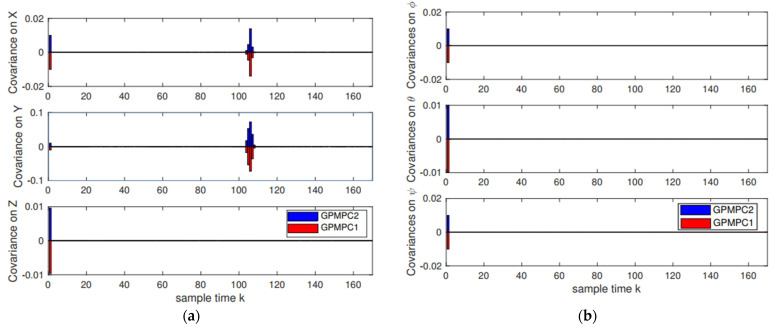
Covariance results: (**a**) Positions covariance; (**b**) Attitudes covariance.

**Figure 13 sensors-22-07887-f013:**
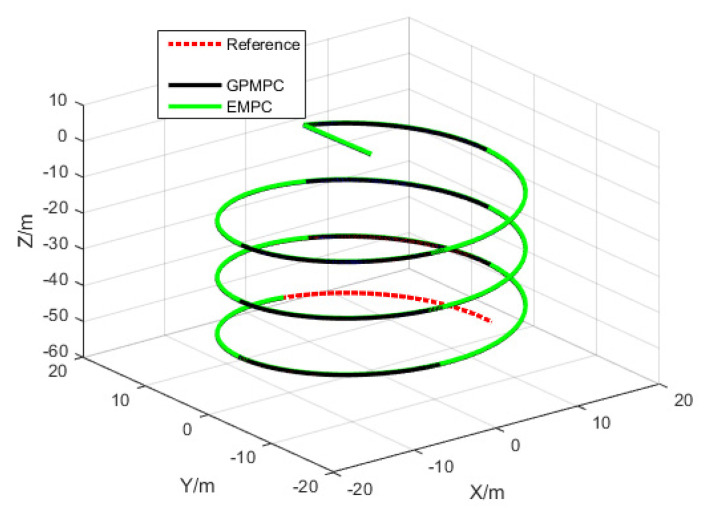
Elliptical trajectory tracking.

**Figure 14 sensors-22-07887-f014:**
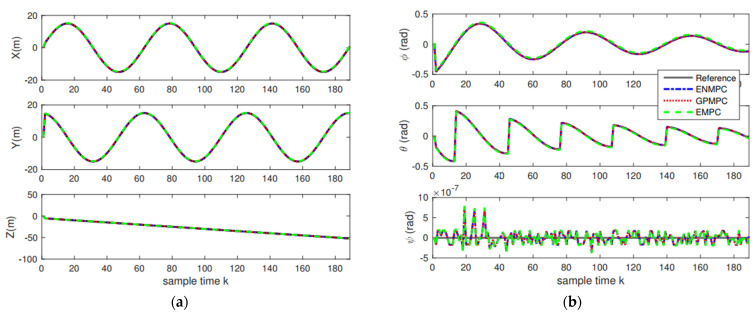
Tracking results: (**a**) Positions tracking; (**b**) Attitudes tracking.

**Figure 15 sensors-22-07887-f015:**
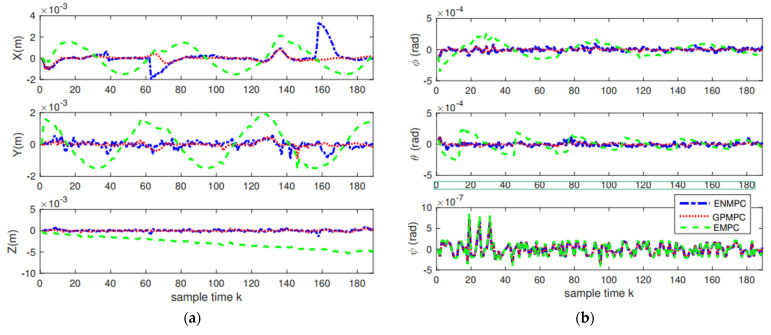
Tracking errors: (**a**) Positions errors; (**b**) Attitudes errors.

**Figure 16 sensors-22-07887-f016:**
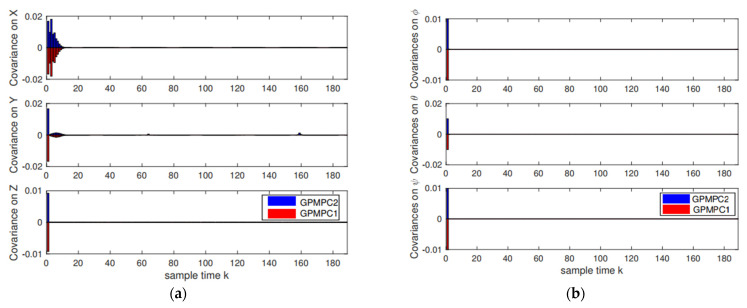
Covariance results: (**a**) Positions covariance; (**b**) Attitudes covariance.

**Table 1 sensors-22-07887-t001:** Mean absolute error.

Controller	ϕ	θ	ψ	$\dot{\emptyset }$	$\dot{\theta }$	$\dot{\psi }$
CT	0.0077	0.0284	0.0488	0.0135	0.0239	0.0289
Adaptive	0.0216	0.1226	0.1009	0.0378	0.0451	0.1969
The Proposed PD	0.0209	0.0205	0.1392	0.0072	0.0137	0.0349

**Table 2 sensors-22-07887-t002:** Training errors and tracking errors.

**Training Methods\Training Errors**	**Positions**	**Attitudes**
GPMPC1	4.3496 × 10^−4^	4.3030 × 10^−9^
GPMPC2	4.3618 × 10^−4^	1.5746 × 10^−8^
**Controllers\Mean Absolute Errors**	**Positions**	**Attitudes**
EMPC	0.0287	9.6239 × 10^−11^
ENMPC	0.0050	7.6412 × 10^−11^
GPMPC	0.0049	2.2407 × 10^−11^

**Table 3 sensors-22-07887-t003:** Training errors and tracking errors.

**Training Methods\Training Errors**	**Positions**	**Attitudes**
GPMPC1	6.1494 × 10^−8^	3.8282 × 10^−10^
GPMPC2	5.1161 × 10^−7^	1.1889 × 10^−9^
**Controllers\Mean Absolute Errors**	**Positions**	**Attitudes**
EMPC	0.0287	0.0263
ENMPC	0.0050	1.8769 × 10^−4^
GPMPC	0.0049	8.4033 × 10^−5^

## Data Availability

Not applicable.

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
