# Peer review of "Trajectory Modeling by Distributed Gaussian Processes in Multiagent Systems"

_sensors, 2022, doi:10.3390/s22207887_

Round 1

Reviewer 1 Report

Dear authors;

After reading the manuscript, I think that this topic is a good issue. I have several comments like as:

- Why there are (M-1) neighbors in lines (342-343), please talk clearly the parameter.

- The author should give block schematic of controlling data-driven model,

- The figure 1 of consensus/fusion flowchart should be described more and more.

- Authors should explain about unmanned quadrotor controlling variables, which is dataset for training of proposed algorithm. 

- How to define the parameters of GP for controling.

- In Definition 2 (Probability space) should be adjusted from Chinese (R space) to English (line 209)

- The training procedure of the proposal distributed GP model of 6 joints (Figure 7) has not achieved convergence as well as good results, so why you stop this learning procedure to get controlling parameters.

- Authors should explain more why choose 2 plants as manipulator and quadrotor for experiments to evaluate the effectiveness of the proposed algorithm.

Above, I have gave you several comments, I hope you will check and adjust accordingly.   

Reviewer 2 Report

This paper proposed a distributed GP based model learning strategy. For the continuous- and discrete- time system, this paper respectively designed GP based PD controller and GP based MPC controller to address the problem. Finally, the effectiveness of the proposed method is analyzed and verified by two examples. Here are some suggestions for improving the quality of the paper.
1. Linguistics, readability of the paper should be further polished.
2. In the introduction, the description of the current research status and difficulties is not detailed enough. It is suggested to add a description of the previous work to better highlight the contribution of the paper. At the same time, it is suggested to cite some recent articles to highlight the frontier of the article.
3. What’s the limitation of your method? Are there other ways that the results can be further improved?
4. Figure 2 in the paper is not clear enough. A higher definition picture is recommended. Could you please describe Figure 1 and Figure 3?
5. In conclusion part, more future works and challenges are recommended.
6. Please revise the references according to the requirements of journal format.

Reviewer 3 Report

1. In Figure 7, authors indicated that the proposed distributed GP model is better than the other two models. However, authors did not describe the training results between three models. It is difficult for readers to realize which model is more accurate.

2. In Figure 9, authors mentioned that the proposed distributed GP (GPMPC2) was more efficient and better trained. What is the difference between GPMPC1 and GPMPC2. Their results at 90% confidence seem to be very close. Some explanations are required. 

3. In Figure 15(a), the tracking error in the Z direction increases with time. What is the reason causing this result? Authors are suggested to explain this result.

4. The model proposed by this study was verified two particular trajectory tracking examples. Authors have to discuss the applications and limitations of their model. Is the model applicable for more complicated trajectory tracking cases?
